

# Pedagogical control scales of vertical jumping performance in untrained adolescents (13–16 years): research by strata

Santiago Calero-Morales[1], Victor Emilio Villavicencio-Alvarez[1], Elizabeth Flores-Abad[2] and Antonio Jesús Monroy-Antón[3]

[1] Department of Human and Social Sciences, Universidad de las Fuerzas Armadas-ESPE, Quito, Sangolquí/Pichincha, Ecuador
[2] Universidad de Ciencias de la Cultura Física y el Deporte "Manuel Fajardo", Havana, Cuba
[3] ESERP Business & Law School, Madrid, Spain

Corresponding author
Santiago Calero-Morales,
scmvoley@gmail.com

## ABSTRACT

**Background:** A scale is used to establish performance ranges in different sciences, it being necessary to design specialized biological and pedagogical indicators in physical activity, sport and health.

**Objective:** To design a scale for the pedagogical control of the vertical jumping ability in untrained adolescents (13–16 years), stratifying the sample by age range, ethnicity, urban and rural area, socioeconomic level, and gender.

**Methods:** A representative sample of the Ecuadorian population ($n = 3,705$) is studied, classifying it into the aforementioned strata, controlling the vertical jump by ISAK I and II level experts, applying the Sargent Test to measure vertical jumps on a multi-force wall, establishing scales with seven percentile levels, and making comparisons related to chronological age, gender, socioeconomic, and genetic indicators.

**Results:** Significant differences in the vertical jumping performance were determined according to the category or age range (13–14 ≠ 15–16 years) and by gender (w = 0.000). Various levels of performance were determined, classifying the maximum level as talented in the female gender (≥40 cm; and ≥42 cm) and male gender (≥47 cm; and ≥57 cm) in the 13-14 and 15-16 years categories, respectively. Sampling comparisons by geographical area only determined significant differences in the male gender, with the jumping ability being higher in urban areas (13-14 years: w = 0.046; 15-16 years: w = 0.013). The comparison by ethnic groups showed significant differences (k = 0.030), favoring the Afro-Ecuadorian ethnic group in both genders, while there are significant differences by socioeconomic level, especially between the middle and lower classes.

**Conclusions:** The present research solves the lack of a tool for making correct didactic decisions related to the vertical jumping ability, taking into account various important stratified indicators. The complementary conclusions show significant differences according to the category stratum or age range, the gender stratum, and the ethnic stratum in females and males, where the best average rank favored the Afro-Ecuadorian ethnic group in both genders. There are significant differences in

the geographical area stratum in the male gender, and differences in the socioeconomic stratum in favor of the upper and middle classes.

## INTRODUCTION

The improvement of basic physical skills is one of the essential contents addressed by the physical activity sciences and sport, including other applied disciplines such as medical sciences (*Wrona et al., 2023*; *Sagarra-Romero et al., 2017*), being also a component managed by the pedagogy that is part of the evaluation of the educational process (*Herrmann et al., 2019*), which provides interpretable and useful results for making the right decisions to optimize the teaching process (*Natriello, 1987*).

The ability to jump improves body schemata in various aspects, given the comprehensive muscle work required in terms of general coordination dynamics (*Aguilera, 2022*), involving recognition and structuring of space and time, where balance and impulse have a significant role (*Montoro-Bombú et al., 2023*; *Morin et al., 2019*). Jumping in compulsory secondary education (ESO) is implicit in a number of sports and pre-sports games (*Santamaría & Frómeta, 2020*; *Betancourt, Quilca & O'farrill, 2020*) and in the baccalaureate and professional training it features in numerous sports; therefore, jumping is also included in physical fitness tests (*Ashley & Kawabata, 2023*; *Zhao et al., 2021*), as it is the basis for future learning and technical moves specific to certain sports (jumping in volleyball and basketball, the high jump and long jump in athletics, *etc.*). The abovementioned context is one of the most commonly used biological and pedagogical justifications for the control of jumping in the different general and specialized physical activity programs.

The development of motor skills favors the mastery of different physical capacities (*Calero-Morales et al., 2023*), including jumping (*Jakšić et al., 2023*; *Hernández et al., 2022*). Jumping is one of the basic physical skills trained directly and indirectly through different curricular activities, being an essential skill in locomotion; therefore, its improvement is implicit in numerous physical education programs (*Hernández et al., 2022*; *Pérez, 2021*) and in the planning processes of numerous cooperation/opposition, combat, and individual sports (*Jakšić et al., 2023*; *Tai et al., 2022*; *Naula & Ayala, 2021*; *Santamaría & Frómeta, 2020*). Jumping is one of the most complex motor skills, involving numerous muscular planes in an integrated way, being a succession of partial and generalized synergistic compensatory attitudes, and of general and specific movements, which make it possible to balance the physical organism when making contact with the floor (*Wallon, 1979*).

Although there are other factors that condition the performance of the jumping ability, such as body weight (*Rendón et al., 2017*) and other components related to genetics (*Beunen et al., 2003*; *Massidda, Scorcu & Calò, 2014*), evaluating the jumping potential as a

physical skill implies relating its potential to physical strength, speed and motor coordination (*Jakšić et al., 2023*; *Burhaein, Ibrahim & Pavlovic, 2020*; *Bishop et al., 2021*). Therefore, its evaluation indirectly describes values related to the three capabilities mentioned, since a jump, being a multi-joint motor action, demands the production of strength combined with power and coordination (*Fuchs et al., 2019*), where the maximum strength ratio is essential to develop the explosiveness of the jump, hence the notable bond between explosive strength and the range that is achieved after a given jump.

The development of jumping ability in the early adolescent stage goes through the third and fourth phases of its evolution (*Cidoncha & Díaz, 2010*) as described by *Sanchez-Bañuelos (1984)*. The third phase (10-13 years) improves the jumping ability through initiation into specific skills and tasks of a playful/sports and recreational nature, although in one way or another always emphasizing the application of sports-type activities, perfecting common generic skills in different sports, and being part of the introduction of specific sports techniques, including the different educational teaching activities to improve the said sports techniques. In the fourth phase of this evolution (14-17 years), the development of specific motor skills is essential as part of the initiation into sports specialization, through technical and tactical training in real game situations.

In their evaluation, physical abilities have a more quantitative and objective character, and although they also have a qualitative character, it is the quantitative aspects that are used as an indicator of excellence, being more objective due to their dependence on the perceptual and motor structure (*Martínez López, 2002*). For this reason, performance assessment tests contain specific exercises that include vertical multi-jumps for 15 s (*Vittori, 1995*) and are hence classified as multi-jump speed tests. Their usefulness is related to their educational nature, which includes the subject's performance, prognosis, classification values, diagnosis, motivation and research.

Control of the jumping ability is essential to establish methodological actions for its improvement, and is understood as the action of measuring and evaluating a certain process. Measuring and evaluating is of paramount importance in the learning/teaching process (*Soares et al., 2020*), including the biological control of sports preparation (*Fukuda, 2018*). Measuring involves the quantitative collection of information under certain conditions, and evaluating involves the identification of information for decision making and feedback.

The scales constructed for physical education and sports programs are essential tools in pedagogical evaluation, widely used in the psychological preparation component (*Behnke et al., 2019*), but limited in other preparation components of proven importance, such as technical/tactical performance components. They are also important tools in the evaluation of physical abilities in untrained individuals, given the lack of scales for the control of physical abilities in the references consulted, with an emphasis on scientifically untrained individuals through the sports training management process.

In the literature there are scales to control basic anthropometric indicators that allow the diagnosis of possible sports talents in different sports, as is the case of *Granja & Frómeta (2018)*. In some articles, a basic scale for the long jump without an impulse race is included for artistic gymnastics athletes (5-8 years old) of both genders, but it is built with

only a small sample (*Palacios, Guerrero & Caicedo, 2020*; *Frómeta, Cuayal & Jácome, 2019*), and there is one also for Olympic wrestling (*Franco Barcia, Caizaluisa Alvarado & Franco Barcia, 2018*) and for the detection and general selection of sports talent in children/adolescents (9-12 years) as a preliminary indicator in a specific region (Ruminahui) of the Republic of Ecuador, which also includes the horizontal jump (*Tipán & Morales, 2018*). All the articles mentioned in this paragraph contribute to detecting possible general and specific sporting talents, but none includes the vertical jump.

Motor performance control makes it possible to establish the necessary values to classify the subject, providing specific reference standards of physical aptitudes depending on previously selected strata, such as those established for the Spanish environment by *Cadenas-Sanchez et al. (2019)*, in terms of gender and age group, and those established for the Republic of Ecuador by *Flores Abad, Arancibia Cid & Calero Morales (2014)*; for these, measuring and evaluating physical abilities such as jumping allows the classification by scales of a population, and based on this, the establishment of decisions related to the teaching-educational process that can pedagogically evaluate the indicators of biological preparation.

In this sense, given the importance of having tools to evaluate the jumping ability that the physical education and sports professional requires, the research purpose is to design a scale for the pedagogical control of the vertical jumping ability in untrained adolescents (13-16 years old), taking into account various strata such as age range, ethnicity, urban and rural area, the socioeconomic level, and gender.

## MATERIALS AND METHODS

### Participants

Using unrestricted random sampling under the *Calero (2003)* formula, a sample that is higher than that representative of the non-athletically trained Ecuadorian population is studied (total population: $n = 1,326,808$), according to *INEC (2019)* between 13-16 years old ($n = 3,705$), which will be stratified by gender (female: $n = 1,761$; male: $n = 1,944$), by age range with the category 13-14 years as female ($n = 907$) and male ($n = 983$) and the category 15-16 years as female ($n = 854$) and male ($n = 961$); ethnic groups are classified according to *INEC (2010)* into Mestiza (female: $n = 1,530$; male: $n = 1,700$), Montubio (female: $n = 41$; male: $n = 48$), Afro-Ecuadorian (female: $n = 40$; male: $n = 48$), Indigenous (female: $n = 30$; male: $n = 36$), White (female: $n = 62$; male: $n = 73$), Black (female: $n = 17$; male: $n = 16$), and Others (female: $n = 41$; male: $n = 23$), as well as the urban population in both genders and age range (female: $n = 1,226$; male: $n = 1,393$), and those from rural areas (female: $n = 535$; male: $n = 651$), and by social stratum or socioeconomic level into Upper Class (female: $n = 21$; male: $n = 18$), Middle Class (female: $n = 357$; male: $n = 380$) and Lower Class (female: $n = 1,383$; male: $n = 1,546$).

This study, with a sample larger than that previously established, allows the estimates to be more precise and with less risk of error (achieving greater statistical power). It guarantees a rigorous control of the records and the information processing to avoid bias in the measurements, an aspect achieved through supervision by two ISAK level II and III

experts, and prior improvement of the teaching by the professionals involved in the measurements.

Adolescents study in numerous schools in the different provinces of Ecuador, in private, subsidized and public schools, but this was not taken into account in the research on the said stratum as an indicator of sample classification. In the sample selection, the following inclusion criteria were taken into account: a) in the age range studied (13–16 years), this being the characteristic range of the ESO level; b) no presently acquired or recent health problems that affect the research, nor disability problems, nor malnutrition, since subjects with such characteristics require specific curricular adaptations (*Torres et al., 2017*); c) parents/teachers authorize the adolescents' participation in research voluntarily and anonymously, including informed consent; d) have the consent of the adolescents studied; e) adolescents cannot have had systematic training in any sport, but participated only in school physical education and/or recreation programs.

## Instruments

To fulfill the research objective, the Sargent Jump test was used, which determines the vertical jump power (PSV) and is recommended for ages 10–18 years. The classification of the mentioned test and its methodological steps are listed below:

– PSV: the test is used to measure the maximum capacity of the extensor muscles of the lower limbs, aimed at improving basic and specific motor activities for physical education and high-intensity sports. It is developed from a maximum vertical jump with leg and arm impulse, obtaining its final value as the difference of the distance between the reach height of the arm and the height of the jumping, with the arm extended at the moment of standing, measuring the take-off distance in centimeters, and registering the best score out of three attempts.

– The implements to be used in the test are: a multi-strength wall marked in centimeters, and a previously designed sheet for recording the information of the three attempts.

– The evaluations were carried out under identical objective conditions (classification of subjects by strata, classes types, environmental and spatial situations), prioritizing the morning hours (09:00 to 11:00 h) to avoid unfavorable weather conditions (sun, rain characteristic of Ecuador) that might affect the implementation protocol. Mondays were preferred to implement the assessment tests, so that the subjects participating have a more uniform rest over the weekend.

– In the case of the socioeconomic level classification or social stratum, income, education and family employment were taken into account according to the threshold methodology followed by the *INEC (2010)*, where the upper class reaches a range of 845.1–1,000 points (A), the middle class 535.1–845 points (B and C+), and the lower class 0–535 points (C− and D). In the case of classification by ethnicity, effective self-identification is used, in the classification by age range, urban and rural area, and gender, the national identity document and confirmatory verbal exchange are used.

## Procedures

In order to ensure reliability in the records and information processing, professionals in Physical Activity and Sports Sciences of the Republic of Ecuador were accredited with the ISAK I and II techniques (*Esparza Ros, Vaquero-Cristóbal & Marfell-Jones, 2019*), being equally endorsed and granted scholarships by the Guayaquil University with a simultaneous postgraduate course of 96 teaching hours, with the professionals who approved the course collecting the data.

Another fundamental step was obtaining the necessary permissions from the school authorities to carry out the research, informing parents/tutors of the intervention process in terms of its limitations and strengths, and on the instruments to be used. The parents/tutors authorized the adolescents' participation in the research voluntarily, with the necessary respect for anonymity, providing informed consent in compliance with the guidelines of the Declaration of Helsinki.

Randomly, the evaluations ($n = 3,705$) carried out by the physical activity and sports professionals were audited by two independent ISAK II and III level experts, validating a representative sample ($n \approx 348$), to determine the concordance index between the records made with the Sargent Jump test through the Krippendorff's alpha, this being evidence of the reliability of the data record, and an indicator of agreement between observers. This research has been approved by the Ethics Committee of the Universidad de las Fuerzas Armadas-ESPE (2016-104-ESPE-d).

## Data analysis

The data were subjected to the Kolmogorov–Smirnov Test, which determined the non-existence of a normal distribution; therefore, non-parametric statistics were selected to establish the following correlations by strata described below:

1) For all the indicated strata, including all the subjects studied: percentiles 5 (Deficient); 15 (Insufficient); 25 (Regular); 50 (Good); 75 (Excellent); 85 (Outstanding); and 95 (Talented).

2) Gender; age range; urban and rural area; and a comparison between the Upper/Middle/Lower Class in the feminine and masculine gender: Mann–Whitney U test for two independent samples ($w \leq 0.05$).

3) Ethnicity; socioeconomic level: Kruskal–Wallis H test for k independent samples ($k \leq 0.05$).

4) To know the concordance index between the records, Krippendorff's alpha was applied, the evaluation being perfect if it coincides with any of the three previous evaluations carried out by the ISAK professionals with the Sargent Jump test, an acceptable agreement between the evaluators being $\alpha \geq 0.6$.

For the data tabulation Microsoft Excel 2021 (Redmond, WA, USA) was used, and SPSS v25 for Windows as the statistical processor (version 25; IBM, Chicago, IL, USA). The research is declared to be descriptive/explanatory of a correlational order.

## RESULTS

The research studied 3,705 untrained adolescents, with a representative sample audited by two independent experts, and determined an acceptable reliability index in the evaluations, according to Krippendorff's alpha ($\alpha$ = 0.7603).

Table 1 shows the fundamental data of the research, determining the percentiles that evaluate the vertical jump ability in the female gender aged 13-14 and 15-16 years old, and in the male gender of the same age ranges.

For the female gender category of 13-14 years, the adolescents with the highest performance were evaluated with the maximum level of talent, their jumping being greater than or equal to 40 cm, and the minimum of less than or equal to 15 cm (Deficient), and in the 15-16 years category with a maximum value of 42 cm, and a minimum of 15.18 cm. In the case of the male gender category of 13-14 years, the maximum value (Talented) must be greater than or equal to 47 cm, and the minimum less than or equal to 17 cm (Deficient), and in the 15-16 years category the maximum value must be greater than or equal to 57 cm, and the minimum value less than or equal to 20.20 cm. The remaining scales describe intermediate values of a higher or lower level, which are important to classify the adolescents in more detail.

Table 2 justifies the construction of scales by age ranges or training category, by determining significant differences between the vertical jump level of the female and male gender in the 13–14-year-old category (w = 0.000), and in the 15-16 years category (w = 0.000), in both cases in favor of the male gender, which presents higher average ranges (AR).

On the other hand, Table 3 also justifies the classification by age ranges of the sample studied, determining significant differences between the 13-14- and 15-16-year-old females (w = 0.000), as well as the 13-14- and 15-16-year-old males (w = 0.000).

Table 4 shows the comparison of the urban and rural area stratum, with no significant differences in the female gender in the 13–14-year-old category (w = 0.584) and the 15–16-year-old category (w = 0.697). However, in the male gender there were significant differences in favor of the urban area (13-14 years: w = 0.046; 15-16 years: w = 0.013), as there is a higher average range, representative of a higher vertical jump performance.

In Table 5, the comparison of the different ethnic groups that live in Ecuador presents significant differences, both for females (k = 0.012) and for males (k = 0.030), with the Afro-Ecuadorian ethnic group having the highest level of vertical jump ability in both the female gender (AR: 1,154.95) and the male gender (AR: 1,152.94), as established with the average ranges of the Kruskal-Wallis H test.

In the case of the socioeconomic level, Table 6 indicates significant differences in the socioeconomic level of the three social strata analyzed, both in females (k = 0.000) and in males (k = 0.003). In terms of average ranks, the upper class presents a better score in the female rank (983.74), and there is a better average rank in the middle class for males (1,061.70).

However, the comparison with the Mann-Whitney U Test shows the non-existence of significant differences between the upper class and the middle class (Table 7), in both the

**Table 1 Percentiles.**

**Statisticians**

|  |  | Female 13–14 | Female 15–16 | Male 13–14 | Male 15–16 |
|---|---|---|---|---|---|
| N | Valid | 907 | 854 | 983 | 961 |
| Average |  | 27.17 | 28.23 | 30.49 | 38.25 |
| percentiles | 5 ≤ Deficient | 15.00 | 15.18 | 17.00 | 20.20 |
|  | 15 ≥ Insufficient | 20.70 | 21.50 | 21.20 | 25.30 |
|  | 25 ≥ Regular | 22.00 | 22.90 | 23.00 | 27.65 |
|  | 50 ≥ Good | 26.00 | 28.00 | 30.00 | 39.00 |
|  | 75 ≥ Excellent | 32.00 | 33.00 | 37.00 | 47.00 |
|  | 85 ≥ Outstanding | 34.50 | 36.00 | 40.50 | 50.00 |
|  | 95 ≥ Talent | 40.00 | 42.00 | 47.00 | 57.00 |

**Table 2 Comparison by gender.** Mann-Whitney U test.

**Ranks**

|  |  | Gender.ALL13–14, Strata | | | Gender.ALL15–16, Strata | | |
|---|---|---|---|---|---|---|---|
|  |  | N | Mean range | Sum of ranks | N | Mean range | Sum of ranks |
| Categories | Female | 907 | 838.46 | 760,485 | 854 | 660.1 | 563,729.5 |
|  | Male | 983 | 1,044.26 | 1,026,510 | 961 | 1,128.29 | 1,084,290.5 |
|  | Total | 1,890 |  |  | 1,815 |  |  |

**Test statistics**

|  | Gender.ALL13–14 |  | Gender.ALL13–14 |
|---|---|---|---|
| Mann–Whitney U | 348,707 | Mann–Whitney U | 198,644.5 |
| W Wilcoxon | 760,485 | W Wilcoxon | 563,729.5 |
| Z | −8.192 | Z | −18.999 |
| Asymptotic sign (bilateral) | 0.000 | Asymptotic sign (bilateral) | 0.000 |

female gender (w = 0.956) and male (w = 0.589), although there are significant differences between the middle and lower classes in both genders (female: w = 0.000; male: $p$ = 0.001). The above shows statistically that the vertical jump power is similar in the upper and middle socioeconomic strata, but in the lower strata the jumping ability is significantly lower.

## DISCUSSION

Taking into account the results achieved, a scale has been designed for the pedagogical control of the vertical jumping ability in untrained adolescents (13-16 years), taking into account various strata such as age range, ethnicity, urban and rural area, socioeconomic level, and gender.

**Table 3 Comparison by categories or age ranges.** Mann-Whitney U test.

**Ranks**

| Female.Strata | | N | Mean range | Sum of ranks | Male.Strata | | N | Mean range | Sum of ranks |
|---|---|---|---|---|---|---|---|---|---|
| Category/Range | Female 13–14 | 907 | 832.76 | 755,311.00 | Category/Range | Male 13–14 | 983 | 778.51 | 765,271.00 |
| | Female 15–16 | 854 | 932.24 | 796,130.00 | | Male 15–16 | 961 | 1,170.94 | 1,125,269.00 |
| | Total | 1,761 | | | | Total | 1,944 | | |

**Test statistics**

| | Female | Male |
|---|---|---|
| Mann–Whitney U | 343,533 | 281,635 |
| W Wilcoxon | 755,311 | 765,271 |
| Z | −4.104051 | −15.413 |
| Asymptotic sign (bilateral) | 0.000 | 0.000 |

**Table 4 Comparison of urban and rural stratum.** Mann-Whitney U test.

**Ranks**

| | | Female 13–14, Strata | | | Male 13–14, Strata | | | Female 15–16, Strata | | | Male 15–16, Strata | | |
|---|---|---|---|---|---|---|---|---|---|---|---|---|---|
| | | N | Mean range | Sum of ranks | N | Mean range | Sum of ranks | N | Mean range | Sum of ranks | N | Mean range | Sum of ranks |
| Zone | Urban | 643 | 450.95 | 289,960 | 699 | 503.50 | 351,949 | 583 | 429.74 | 250,539.5 | 694 | 494.73 | 343,341 |
| | Rural | 264 | 461.43 | 121,818 | 284 | 463.69 | 131,687 | 271 | 422.68 | 114,545.5 | 267 | 445.32 | 118,900 |
| | Total | 907 | | | 983 | | | 854 | | | 961 | | |

**Test statistics**

| | Female 13–14 | Male 13–14 | Female 15–16 | Male 15–16 |
|---|---|---|---|---|
| Mann–Whitney U | 82,914 | 91,217 | 77,689.5 | 83,122 |
| W Wilcoxon | 289,960 | 131,687 | 114,545.5 | 118,900 |
| Z | −0.547588 | −1.993 | −0.390 | −2.472 |
| Asymptotic sign (bilateral) | 0.584 | 0.046 | 0.697 | 0.013 |

In the international literature, it is common to establish comparisons in various performance indicators taking gender into account, where notable differences are normally determined in various abilities and physical capacities (*Espinosa-Albuja, Haro-Simbaña & Calero, 2023*), including jumping ability, for which different control tables are established, emphasized specifically in the sports area (*Granja & Frómeta, 2018*; *Palacios, Guerrero & Caicedo, 2020*; *Frómeta, Cuayal & Jácome, 2019*; *Franco Barcia, Caizaluisa Alvarado & Franco Barcia, 2018*; *Tipán & Morales, 2018*).

The differences established by gender in the present research are notable, an aspect that justifies their math differentiation, as well as the differentiation by age ranges or sports categories, where *Cidoncha & Díaz (2010)* classify the evolution of jumping ability by

**Table 5 Ethnic comparison.** Kruskal–Wallis H test.

**Ranks**

| Ethnicity | | N | Mean range | Ethnicity | | N | Mean range |
|---|---|---|---|---|---|---|---|
| Female | Mestizo | 1,530 | 871.46 | Male | Mestizo | 1,700 | 969.59 |
| | Montubio | 41 | 939.17 | | Montubio | 48 | 1,103.52 |
| | Afro-Ecuadorian | 40 | 1,154.95 | | Afro-Ecuadorian | 48 | 1,152.94 |
| | Black | 17 | 833.47 | | Black | 16 | 643.69 |
| | Aboriginal | 30 | 979.90 | | Aboriginal | 36 | 979.72 |
| | White | 62 | 933.40 | | White | 73 | 923.69 |
| | Other | 41 | 779.50 | | Other | 23 | 909.76 |
| | Total | 1,761 | | | Total | 1,944 | |

**Test statistics**

| | Ethnicity female | Ethnicity male |
|---|---|---|
| Kruskal–Wallis H | 16.269 | 13.959 |
| gl | 6 | 6 |
| Asymptotic sig. | 0.012 | 0.030 |

**Table 6 Comparison by socioeconomic level.** Kruskal–Wallis H test.

**Ranks**

| Groups.Class.Female | | N | Mean range | Groups.Class.Male | | N | Mean range |
|---|---|---|---|---|---|---|---|
| Female | Upper class | 21 | 983.74 | Male | Upper class | 18 | 979.61 |
| | Middle class | 357 | 980.84 | | Middle class | 380 | 1,061.70 |
| | Lower class | 1,383 | 853.67 | | Lower class | 1,546 | 950.49 |
| | Total | 1,761 | | | Total | 1,944 | |

**Test statistics**

| | Female | | Male |
|---|---|---|---|
| Kruskal–Wallis H | 18.624 | Kruskal–Wallis H | 11.978 |
| gl | 2 | gl | 2 |
| Asymptotic sig. | 0.000 | Asymptotic sig. | 0.003 |

phases of biological development, which includes a third phase for adolescents aged 10–13 years, and a fourth phase for adolescents aged 14–17 years (*Sanchez-Bañuelos, 1984*), it being logical that differences in chronological and biological age imply different stages of motor development. Therefore, the assessment tests will indicate different degrees of performance, as evidenced in the results of *Sinkovic et al. (2023)*, where the speed capacities, agility and explosive power depend on the biological age, and imply different levels of jumping performance, and performances that derive from greater or lesser advantages in the tests according to the indices of biological maturity, even in the same age group (*Malina et al., 2015*).

**Table 7 Comparison by socioeconomic level.** Mann–Whitney U test.

| Female | N | Mean range | Sum of ranks | Male | N | Mean range | Sum of ranks |
|---|---|---|---|---|---|---|---|
| Upper class | 21 | 188.21 | 3,952.50 | Upper class | 18 | 185.19 | 3,333.50 |
| Middle class | 357 | 189.58 | 67,678.50 | Middle class | 380 | 200.18 | 76,067.50 |
| Total | 378 | | | Total | 398 | | |
| **Female** | **N** | **Mean range** | **Sum of ranks** | **Male** | **N** | **Mean range** | **Sum of ranks** |
| Middle class | 357 | 970.26 | 346,383.00 | Middle class | 380 | 1,052.02 | 399,768.00 |
| Lower class | 1,383 | 844.75 | 1,168,287.00 | Lower class | 1,546 | 941.74 | 1,455,933.00 |
| Total | 1,740 | | | Total | 1,926 | | |

**Test statistics**

| | Female. Upper class-Middle class | Male. Upper class-Middle class | Female. Middle class-Lower class | Male, Middle class-Lower class |
|---|---|---|---|---|
| Mann–Whitney U | 3,721.500 | 3,162.500 | 211,251.000 | 260,102.000 |
| W Wilcoxon | 3,952.500 | 3,333.500 | 1,168,287.000 | 1,455,933.000 |
| Z | −0.055 | −0.540 | −4.209 | −3.464 |
| Asymptotic sig. | 0.956 | 0.589 | 0.000 | 0.001 |

The differences between gender and age range are partly anatomically and functionally justified given the existing differences in the body structure of adolescents aged 11–16 years, as well as in other younger and older age ranges (*Kirchengast, 2010*). Even the biological development in the same age group and gender differs significantly in terms of physical condition (*Sinkovic et al., 2023*; *Sitovskyi et al., 2019*), being notably different in the vertical jump in untrained individuals, as demonstrated in the present research. This justifies the construction of different pedagogical tools for the control of physical preparation according to the training category or age range, and gender.

Regarding the influence of ethnicity on sports performance in key indicators such as speed and resistance, *Pitsiladis (2011)* highlights that the genetic diversity described in the literature cannot be excluded from the analyses, but so far environmental factors are the ones that most affect sports results (*Cerit, Dalip & Yildirim, 2020*), although authors such as *Massidda, Scorcu & Calò (2014)* specify that the creation of a genetic model can predict the variability of athletic performance, including the vertical jump, taking into account genetic variations.

In this sense, sociocultural and economic conditions could have marked influences on the jumping ability performance in untrained individuals in the adolescent stage, taking into account that the greatest percentage of Afro-Ecuadorians live in Esmeraldas province (*DPE, 2012*), one of the poorest in Ecuador, but one of the most relatively active in terms of specialized physical activity and resilience (*Mina-Barahona & Proaño-Mina, 2022*), which would partly explain the higher performance in the vertical jumping ability than the rest of the ethnic groups studied. In this sense, in terms of search and sports selection, the Afro-Ecuadorian ethnic group has the best indicators in the vertical jumping ability, a

useful indicator in various sports where jumping is a determining variable of sports performance.

Another sociocultural and economic aspect that would probably explain the absence of significant differences in the female gender in the vertical jumping ability between urban and rural areas, while in the male gender there are significant differences by area, are the roles adopted by each gender in Latin America and Ecuador (*Ramirez, Manosalvas & Cardenas, 2019*), which are eminently unfavorable in terms of active physical activity in Ecuadorian females, regardless of the geographical areas where they live, as well as the superior development of physical/sports and recreational infrastructure existing in urban areas, which favors a more active physical activity in the area of physical education and sport.

The previous idea is based on the fact that socioeconomic conditions determine the systematic practice of physical activity from school onwards, in favor of the high/medium purchasing levels. Therefore, according to *Powell, Slater & Chaloupka (2004)*, the availability of environmental factors favors physical activity and sports, having a relationship with minority ethnic groups in addition to the socioeconomic factor, as well as the logistical support of parents (*Raudsepp, 2006*), given that adolescents with a higher socioeconomic level are more physically active (*Stalsberg & Pedersen, 2010*), and all this favors a better level of physical abilities, as is evident in the present research, where the lower class presents lower performance indicators in vertical jumping compared to the rest of the socioeconomic classes analyzed.

Taking into account the above and the significant differences obtained, it was considered appropriate to create scales for the pedagogical control of the vertical jumping ability that consider exclusively the gender strata and age range (13–14 and 15–16 years). This was despite the fact that the research considers it necessary to take into account other strata of analysis such as ethnicity and the socioeconomic level, which should be considered for more specific research in future, where a more complete and comprehensive study is required.

## Strengths and limitations

The main strength of the research is the originality of the subject, due to the lack of scales for the pedagogical control of the vertical jumping ability in untrained adolescents, with emphasis on the age range studied, according to the consultation carried out in the different primary sources of research. Another relevant aspect is that the study was carried out with a representative sample of the untrained population, allowing generalization of the results in physical education programs and the sports search and selection process, respecting international reliability indicators. In addition, the classification of the sample by strata, with an emphasis on indicators of ethnicity and socioeconomic class, lays the foundations for other directly related research.

However, considering that the ethnic stratum is classified by self-perception criteria, this could be based on a split self-concept, where the reality might be different from what was perceived by the adolescent studied. Therefore, in the opinion of the research authors,

caution must be exercised in the data analysis in the ethnic component, as there is a need to carry out new studies to better classify the sample under study through genetic tests.

## CONCLUSIONS

The research shows significant differences according to the category stratum or age range, the gender stratum, and the ethnic stratum in females and males, where the best average range was demonstrated by the Afro-Ecuadorian ethnic group in both genders. In the case of the area stratum comparisons, there are no differences in the female gender in the urban or rural area, either in the 13–14-year-old category or in the 15–16-year-old category, although in the male gender there were significant differences. There are also differences in the vertical jumping ability according to the socioeconomic stratum, favoring the upper and middle classes, the possible causes being the subject of new studies. The foregoing justifies theoretically and methodologically the creation of performance scales for the vertical jumping ability, taking into account the differences generated by various of the analysis strata as they are constructed in the present research, even demonstrating the need to study other physical abilities (including jumping) in the same categories, and in other age ranges.

### Funding
This research was funded by Universidad de las Fuerzas Armadas-ESPE (2016-104-ESPE-d), and the Universidad de Guayaquil-Oficina de Proyectos Rentables (FCI-2014). The funders had no role in study design, data collection and analysis, decision to publish, or preparation of the manuscript.

### Grant Disclosures
The following grant information was disclosed by the authors:
Universidad de las Fuerzas Armadas-ESPE: 2016-104-ESPE-d.
Universidad de Guayaquil-Oficina de Proyectos Rentables: FCI-2014.

### Competing Interests
The authors declare that they have no competing interests.

### Author Contributions
- Santiago Calero-Morales conceived and designed the experiments, performed the experiments, analyzed the data, prepared figures and/or tables, authored or reviewed drafts of the article, and approved the final draft.
- Victor Emilio Villavicencio-Alvarez conceived and designed the experiments, prepared figures and/or tables, and approved the final draft.
- Elizabeth Flores-Abad performed the experiments, analyzed the data, prepared figures and/or tables, and approved the final draft.
- Antonio Jesús Monroy-Antón performed the experiments, analyzed the data, authored or reviewed drafts of the article, and approved the final draft.

## Human Ethics

The following information was supplied relating to ethical approvals (*i.e.*, approving body and any reference numbers):

Universidad de las Fuerzas Armadas-ESPE (2016-104-ESPE-d).

## Data Availability

The data is available in the Supplemental File.

## Supplemental Information

Supplemental information for this article can be found online at http://dx.doi.org/10.7717/peerj.17298#supplemental-information.

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
