# Peer review of "Pedagogical control scales of vertical jumping performance in untrained adolescents (13–16 years): research by strata"

_PeerJ, doi:10.7717/peerj.17298_

## Round 0.1 · original submission · Major Revisions

Please take into account the suggestions of our reviewers.

**Language Note:** The review process has identified that the English language must be improved. PeerJ can provide language editing services - please contact us at copyediting@peerj.com for pricing (be sure to provide your manuscript number and title). Alternatively, you should make your own arrangements to improve the language quality and provide details in your response letter. – PeerJ Staff

Reviewer 1 ·

Basic reporting

See Additional Comments

Experimental design

See Additional Comments

Validity of the findings

See Additional Comments

Additional comments

[1] Please, correct medical (line 46).
[2] Line 51-52: The sentence needs to be clarified. Please, adjust.
[3] Line 80-89: This paragraph is redundant with the first. Please, delete it.
[4] Line 110-117: Please mention the contribution of jumping performance to these scales.
[5] Line 126-130: The sentence seems to be contradictory. On the one hand, jumping is important for sports, but the scale was developed in untrained adolescents. Please, correct the sentence.
[6] The authors have a large sample. Did you base the decision on non-parametric statistics considering the K-S test? Why?
[7] Line 138: Talent needs to be corrected for untrained adolescents. Please, adjust.
[8] Why are the authors creating percentiles and then testing the differences? Authors should test differences and create percentiles for jumping performance based on age, sex and socio-economic status.
[9] The tables presentation should be improved.
[10] The first paragraph should indicate the main findings of the current study.
[11] The conclusions should not include statistical values and must be reflective.

Reviewer 2 ·

Basic reporting

The study aims to design a scale for the pedagogical control of vertical jumping ability in untrained adolescents (13-16 years), stratifying the sample by age range, ethnicity, urban and rural area, socioeconomic level, and gender. While the paper is well-structured and follows journal guidelines, It would like the authors to check English language typography before resubmission and respond to the following questions and comments.
Título
Line 1-3: Por favor comprueba si no es más conveniente capacidad de salto vertical.
Line 1-3: Aunque no es relevante, puedes considerar eliminar del título contextualizar en participantes. Ya declaras que son adolescentes no entrenados.

Abstract
Line 21 -23 I recommend revising this sentence; it presents a speculative tone. Furthermore, you state that it is in different sciences and then state that it is specialised in physical activity and sports. An alternative could be: In physical activity and sports, a scale is about providing a set of statistically predetermined outcomes...; OR.... A scale is a statistically tested standard that is established for the purpose of standardisation......
Line 23-25 please check the need to keep this sentence: tratifying the sample by age range, ethnicity, urban and rural area, socioeconomic level, and gender...
Line 25-29: I recommend stating in summary the procedure, e.g. how vertical jumping ability was measured, at what time it was measured, which instruments were used for the measurement, which parameter was assessed (I infer jump height).
Line 29 -38: State the results in a concrete way. Example: significant differences were found on _____ and ____ (w=___) and on ____. Possible are identified according to the following jump heights ______ etc.
Line 38-40: this is a good introduction for summary, but I doubt it is a conclusion. EXAMPLE: it is concluded that Afro-Ecuadorians jump more or less; it is concluded that jump ranges are between ___ and ___. It is concluded that to be talented, you need to jump between ____ and ____.

Instruction.
Line 45: Basic physical capacity or Basic motor skills, I don't understand why basic physical skills.
Line 46: What other disciplines?
Line 45-47: I don't understand how this reference is contextualised, I can't find it in the cited studies.
Line 53-54: I recommend you to eliminate these speculations.
Line 63: which ones?
Line 64: evaluate or measure?
Line 68-40: this is not real. In the vertical jump the maximum force production is not the determinant of the result. Here it is the momentum that plays a key role (see Stewart 2022 or Samozino 2019).
Line 68 - 130. I think it is important to reorganise the introduction and only put what is important.

Experimental design

Methodology

Remember to write in the past tense. This is a criticism that extends throughout the methodology. Example: was studied.
Describe ages with mean ± SD
It is necessary to state anthropometric data as required by the journal, height, body mass (weight) and body mass index.
Since the data are with minors, it is necessary to send a copy of the informed consent form signed by the parents.
Line 35: Please describe in text the results of the formula and how you determine that the sample is larger than representative. I find it difficult to understand this sentence if it is not justified. I recommend strengthening the sample size by statistical power.
Line 137: this is the total untrained population aged 13-16 years? I suspect it is not the total population of Ecuador because it would not be statistically correct.
Line 148: this sentence is repeated.
Line 149-152- This idea seems to me to be purely speculative. These ISAK level II and III experts are specialised in anthropometry, but not in measuring vertical jump, this type of measurement is not within their field of study.
Line 160-161: they are minors, the consent is signed by their parents. Be careful with this.
Line 153 -162: and the exclusion criteria?
Line 156-189: there is a big mix between procedure and instruments. State here only which instruments were used.
Line 165-167: Please review Samozino in his study (when the vertical jump does not determine the power of the lower limbs), here I recommend to state only: to determine the height of the jump ... The test ..... was used. And put the reference
Line 168- 169: is this real? or is it only the height of the jump? only high intensity sports?
Line 170 -174: no no no, please there are different criteria to perform this same jump, make a detailed description and position yourself with an author. Example how was the leg separation, allowed angle of impulse, jumping distance from the wall, how was the maximum jumping distance controlled and corrected for scoring, was it the same person as the average person, 11115 jump attempts (3705 x 3) etc.
Line 178: for those who know the equator it is difficult to state this. I consider that this is only possible if all 3705 subjects are brought together to do the tests in a single scenario. Was this what happened? Were the students not prevented from doing any kind of intense physical activity on weekends?
Line 191: state here lines 168 -182.
Line 195: I don't understand why this reference.
Line 193 -197: this paragraph seems to me to be out of context and outside the procedure.
Line 198-207 IDEM
Line 203-205. I find this very interesting, but how was this analysis of standardisation of measurements carried out, did they watch videos, were they present at this number of tests, what were the indicators they checked (I don't think it is auditing, it must be one of the errors in English)?
Line 206-207: this raises a lot of doubts. As far as I understand, this alpha is more used for dichotomous variables for Likert scales. In our case for inter-examiner agreement analysis it is the intraclass correlation index (ICC), am I right?
Line 218-220: The Mann-Whitney U was used to see the differences between what measures? I assume jump height in cm, no? You should state it here.
Line 221-222: IDEM.
Line 223-226: Review the comment above on Krisppendoff's Alpha. Also, I recommend reviewing the results of this reliability value. Here the most important thing is not the significance level, but the % value of inter-rater agreement.
Line 227: IBM SPSS Statistics (version 27; IBM, Chicago, IL).

Validity of the findings

Results.
It would be more appropriate to state the results in a concrete way, starting with the most important and representative findings. In my opinion the text is too long and sometimes with information that has already been repeated before.
Line 232-234. This is not only repetitive but also more appropriate for the discussion and not for the results. Here the Alpha report is more acceptable.
Line 235-237: I do not question the strong data collection, but if the processing conditions for obtaining the results, these data seem to me somewhat overestimated compared to other scientific reports in similar ages and more taking into account that they are untrained subjects and at school age. See (10.3390/ijerph181910446) (PMC7675629) (https://doi.org/10.3390/jfmk8020048)
Line 248-251: SPSS supports whatever you put into it, it would be appropriate to report only what is necessary, here this test by ranges to my understanding does not report any relevant data according to the purpose of the study. If this is not the case, I would like you to justify:
How did the ranked results help in the selection indicators?
What does the average rank mean and what is the information provided by this report?
What practical information does the sum of ranks provide?
What practical information does the sum of ranks provide?
It would be appropriate to redo the tables and leave only one with the Z-values and the p-value (in your case w and k), the remaining data despite being statistics reported by the commonly used programmes are not of great importance for the reporting of the data.
Line 263-278. It would be interesting not only to report k, but also the type of hotposts, here the information is more complete and we would have to know the multiple significance of the results, also here if it would be appropriate to use the benefits of SPSS and declare only one clip the effect size between groups ( ηp2).
On line 148-149 you say that: The study with a larger sample size than the one established allows the estimates to be more precise and with less risk of error, guaranteeing rigorous control of the records and the processing of the information to avoid bias in the measurements, however we did not find the reports of standard error of the mean, important data to support this sentence.
Although you try to provide complete information, I am still trying to understand how these differences between groups contribute to the objective of the research: to design a scale for the pedagogical control of the vertical jump skill in untrained adolescents (13-16 years old) :::::: if you have comments on this, feel free to express them.

Additional comments

Discussion
Line 280: you can download the first paragraph of results for this section.
Line 282-284: check the references because there are some that do not include jump control tables and do not justify their reference. E.g. Granja & Frómeta, 2018, Palacios et al., 2020 etc. Are the differences notable or significant? Please correct.
Although it would be important to establish more points of contradiction with another study and discuss more your results with others previously published, this section is very complete. One point I consider important is to present why your results for these ages are the best found in the published literature.
Conclusions
You can remove the results from the conclusions and present the most important results found.

---

## Round 0.2 · Minor Revisions

Dear author,

in my opinion the paper is a version that is very close to being published.
nerveless, you should introduce in the text the right place where the tables should be inserted.

Additionally, should be formatted according to the best scientific practices for table presentations. current form is like direct import from software outputs.

Best regards

Reviewer 2 ·

Basic reporting

The authors manage to give accepted answers to the comments of the previous review. In this sense, we recommend the publication of the study.

Experimental design

No comments

Validity of the findings

No comments

Additional comments

No comments

---

## Round 0.3 · accepted · Accept

The original Academic Editor is unavailable so I have taken over handling the submission.

There were minor changes required to the manuscript which the authors have addressed. I am pleased to recommend the amended manuscript for publication. Thank you for choosing PeerJ and we look forward to future manuscripts.